

# Causal interactions between ENSO and the North Tropical Atlantic

Thanh Le[1][*] and Deg-Hyo Bae[1][*]

[1]Department of Civil and Environmental Engineering, Sejong University, Seoul 05006, Republic of Korea

[*]Corresponding author(s): Thanh Le (thanhle@sejong.ac.kr) and Deg-Hyo Bae (dhbae@sejong.ac.kr)

**Abstract.** The global climate is impacted by several major climate modes including the North Tropical Atlantic mode (NTA) and the El Niño–Southern Oscillation (ENSO). Although NTA and ENSO are suggested to have connections, there is uncertainty regarding the causal relationship between these climate modes. While previous works focused on the correlation between NTA and ENSO, causal analyses accounting for the influence of other tropical climate modes are lacking. Here we investigate the causal links between ENSO and NTA using outputs from high-resolution climate model simulations and

reanalysis data. Our results suggest robust causal effects of ENSO on NTA and provide insights on the unstable impacts of NTA on ENSO. We observe high consistency between reanalysis data and the models in mimicking the impacts of ENSO on North Tropical Atlantic region. Specifically, most models (14 over 20) and reanalysis data revealed that ENSO is very unlikely to have no causal impacts on NTA. However, there is diverse response of the tropical Pacific to NTA between reanalysis data and the models. While reanalysis data indicates possible impacts of NTA on ENSO and sea surface

temperature over the equatorial Pacific, the majority of models (18 over 20) suggest that the NTA is likely to have no causal effects on ENSO. Hence, the models may underestimate the causal effects of NTA on ENSO, implying that better representation of NTA variability and NTA-ENSO causal connections in the models may improve the predictability of ENSO variations.

**Keywords:** El Niño–Southern Oscillation; high-resolution climate model; North Tropical Atlantic; causal connections;
CMIP6.

## 1 Introduction

The global climate is impacted by major modes of climate variability including the North Tropical Atlantic mode (NTA) and the El Niño–Southern Oscillation (ENSO). The NTA (Enfield and Mayer, 1997; Ham et al., 2013b), typically peaks in boreal spring, is characterized by variations of sea surface temperature (SST) in the north tropical Atlantic region, and has broad

effects on atmospheric circulation and rainfall pattern over the surrounding areas (Chang et al., 2000; Enfield, 1996; Jin and Huo, 2018; Li et al., 2014; Murakami et al., 2018; Saunders and Lea, 2008; Vimont and Kossin, 2007; Wang et al., 2006; Watanabe and Kimoto, 1999). ENSO is a leading mode of global climate variability (Cai et al., 2021; Le and Bae, 2022; Martens et al., 2018; Thirumalai et al., 2017) and exhibits influences on economies and ecosystems worldwide (Cai et al., 2020; Donnelly and Woodruff, 2007; Le et al., 2021; De Luca et al., 2020; McPhaden et al., 2006). Although NTA and





ENSO are suggested to have connections (Cai et al., 2019; Ham et al., 2013b), there is ambiguity regarding the causal link between these climate modes.

In particular, several studies (Alexander et al., 2002; García-Serrano et al., 2017; Saravanan and Chang, 2000; Tokinaga et al., 2019; Zhang et al., 2021) showed a sustained and dominant role of ENSO in the linkage between the tropical Pacific and tropical Atlantic. Other studies (Dommenget et al., 2006; Exarchou et al., 2021; Frauen and Dommenget, 2012; Ham et al.,

2013b) demonstrated the predictive power of tropical Atlantic variability on ENSO. Changes in the Atlantic regions may induce variations in the tropical Pacific and ENSO characteristics (Ham et al., 2013a; Keenlyside et al., 2013; Li et al., 2016; Martín-Rey et al., 2015; Wang, 2019; Wang et al., 2017; Zhang et al., 2019). In addition, there is potential shifts in ENSO-NTA links in the future (Choi et al., 2019; Jia et al., 2019; Liu et al., 2021; Yang et al., 2021). While previous works focused on the correlation between ENSO and NTA, the causal connections between these two modes remain unclear. In addition,

the role of confounding factors of different tropical basins might not be considered in the analyses.

In the present study, we examined the causal interactions between ENSO and NTA using high-resolution model simulations from the Coupled Model Intercomparison Project Phase 6 (CMIP6) and reanalysis data. As warming environment may intensify the regional impacts of NTA and ENSO (Cai et al., 2021; Yang et al., 2021), further understanding of the causal links between these modes is necessary.

## 2    Materials and Methods

### 2.1    Datasets

We employed datasets from the High-Resolution Model Intercomparison Project (HighResMIP; Haarsma et al., 2016) in the CMIP6 (Eyring et al., 2016), which cover the 1950-2014 period. Table S1 lists the models used in this work. Reanalysis data is obtained from the 20th Century Reanalysis Project version 2c (Compo et al., 2011). We utilized monthly data of the

following variables: sea surface temperature, zonal and meridional winds at different levels (i.e., surface, 850-hPa and 250-hPa).

### 2.2    Methods

Based on the approach employed in recent works (Le et al., 2020; Le and Bae, 2019), we calculated the probability of no Granger causality from NTA to ENSO and ENSO to NTA for the period 1950–2014. As we focus on the interactions of

tropical basins (Cai et al., 2019; Chu et al., 2014; Ha et al., 2017; Le et al., 2020; Le and Bae, 2019; Wang, 2019), our analyses considered possible confounding impacts of the Indian Ocean Dipole (IOD) and the Indian Ocean Basin mode (IOB) on the connections between NTA and ENSO. Further details of this approach are given in Section S1. This approach was also used to evaluate the causal influence of NTA and ENSO on the tropical Pacific Ocean and tropical Atlantic Ocean climate.



We computed the indices utilized in this study as follows. The North Tropical Atlantic index (NTA) (Ham et al., 2013b) was given as the average sea surface temperature (SST) anomalies in the boreal spring (March-April-May, MAM) over the North Tropical Atlantic region (90°W–20°E; 0°N–15°N). The ENSO index was calculated as the average SST anomalies in boreal winter (December–January–February, DJF) over the Niño 3.4 region (120–170°W; 5°N–5°S). The IOD index (Saji et al., 1999; Webster et al., 1999) was computed as the difference SST anomalies between two Indian Ocean regions of the western

pole (50–70°E; 10°N–10°S) and southeastern pole (90–110°E; 0°N–10°S) in boreal fall (September–October–November, SON). Lastly, the IOB index (Cai et al., 2019; Chu et al., 2014; Ha et al., 2017) is computed as the first empirical orthogonal function (EOF) of boreal spring (MAM) SST anomalies in the Indian Ocean basin (40°E-120°E, 20°N-20°S).

## 3    Results

Figure 1 describes the normalized time series (i.e., standard deviation $\sigma=1$) of NTA and ENSO in the models (colored
symbols) and the reanalysis data (solid blue line) during the 1950–2014 period. Reanalysis data and the models show consistency in simulating the timing and amplitude of ENSO variations (Figure 1a). Nevertheless, there is divergence between reanalysis data and the models in simulating NTA index (Figure 1b). While the sign of modeled NTA agrees with reanalysis data, the magnitude of NTA is varied across models. For example, model simulations of strong positive NTA events (i.e., index above $+1.5\sigma$, highlighted by cyan shading in Figure 1b) are consistent for the years 1998 and 2010, but
there is divergence for the years 1958 and 1970. Most models underestimate the strength of strong positive NTA events in the years 1958 and 1970. In addition, strong negative NTA events (i.e., lower than $-1.5\sigma$ in Figure 1b) over the years 1974 and 2001 are not well captured by most models. Climate models have biases in the equatorial Atlantic (Richter et al., 2014), though, model simulations still contribute important dataset for evaluating the causal links between NTA and ENSO.

### 3.1    The influence of ENSO on NTA

Figure 2 shows the probability of no Granger causality from ENSO to NTA during the 1950–2014 period. Most model simulations (14 over 20) and reanalysis data indicate that ENSO is very unlikely (i.e., $p$-values were less than 0.1, denoted in yellow bars in Figure 2) to have no causal impacts on NTA. Three models (CAMS-CSM1-0, CMCC-CM2-VHR4, ECMWF-IFS-HR) suggest that ENSO is unlikely (i.e., $p$-values were less than 0.33 and higher than 0.1, highlighted in cyan bars) to have no causal impacts on the NTA. Two models (GFDL-CM4C192 and IPSL-CM6A-LR) had $p$-values from 0.33 to 0.66,
implying that ENSO is as likely as not to have causal impacts on the NTA. The model INM-CM5-H ($p$-value is higher than 0.66) suggests that ENSO is likely to have no causal influences on the NTA. Hence, these results indicate that ENSO has robust causal impacts on NTA during the 1950–2014 period.

Figure 3 describes the response of the north tropical Atlantic SST, surface zonal winds, and surface meridional winds during the boreal spring of year $t+1$ [i.e., MAM($t+1$)] to ENSO of year $t$ [i.e., D($t$)JF($t+1$)]. The multi-model mean shows robust
ENSO impacts on SST over much of the north tropical Atlantic region (Figure 3a), consistent with reanalysis data (Figure





3b). ENSO causal impacts on the surface zonal wind systems are found in the equatorial Atlantic (Figures 3c and d), suggesting possible atmospheric bridge between equatorial Pacific and Atlantic oceans. The effects of ENSO on surface meridional wind systems are seen in the north tropical Atlantic region between 0°N–15°N (Figures 3e and f). We notice high consistency across models for the significant impacts of ENSO on SST and surface wind systems in the tropical Atlantic

region (i.e., regions highlighted by stippling in Figures 3a, c and e).

Figure 4 depicts the outcomes of 20 separate models for the causal effects of ENSO on SST over the tropical Atlantic region. The individual models (Figure 4) exhibit similar pattern compared to the multi-model mean (Figure 3a) and reanalysis data (Figure 3b), implying robust ENSO impacts on NTA SST and high confidence in the models' capability in reproducing ENSO-induced changes in the north tropical Atlantic region.

Further analyses revealed the significant causal impacts of ENSO on the Pacific-Atlantic Walker circulation (Figure 5). As described in Figure 5, ENSO was likely to have causal influences on 850-hPa zonal (Figure 5a) and meridional (Figure 5b) winds over the tropical Atlantic region, consistent with Figures 3c-f. The impacts of ENSO on 250-hPa zonal (Figure 5c) and meridional (Figure 5d) winds further demonstrate robust ENSO signature on convection processes over the tropical Atlantic region. Figure 5e shows that the causal impacts of ENSO on NTA in reanalysis data are consistent on all 41-year sliding

windows, implying the strong response of NTA to ENSO during 1950–2014. The moving causal impacts of ENSO on NTA are varied between models and the $p$-values are higher for the central years from 1976-1989 (Figure 5e). Nevertheless, the models mean $p$-values of all central years are less than 0.33, implying that ENSO is likely to have stable impacts on NTA.

## 3.2    The influence of NTA on ENSO

Figure 6 illustrates the probability for the absence of Granger causality from NTA to ENSO during the 1950–2014 period.

The majority of models (18 over 20) suggest that the NTA is likely ($p$-value is higher than 0.66) to have no causal influences on ENSO. The model IPSL-CM6A-ATM-HR ($p$-values from 0.33 to 0.66) suggests that the NTA is as likely as not to have causal effects on ENSO. Reanalysis data and the model MPI-ESM1-2-XR indicate that the NTA is unlikely (i.e., $p$-values were lower than 0.33 and higher than 0.1) to have no causal impacts on ENSO. There is high consistency across models in suggesting the weak influence of NTA on ENSO. Nevertheless, the results imply that the models underestimate the response

of ENSO to NTA compared to reanalysis data.

The models show high agreement for the nonsignificant impacts of NTA on boreal winter SST, surface zonal and surface meridional winds over the tropical Pacific (Figures 7a, c and e). These impacts are, however, stronger in reanalysis data. Specifically, in reanalysis data, NTA shows significant causal effects on boreal winter SST over equatorial Pacific (Figure 7b). In addition, NTA may induces changes in surface zonal and meridional winds in parts of the tropical Pacific (Figures 7d

and f). The model simulations suggest that NTA exerts impacts on tropical Pacific SST in subsequent boreal summer (Figure S1a) after peaking in boreal spring. However, these impacts weaken in the subsequent fall and winter (Figures S1b and 7b), leading to nonsignificant effects of NTA on ENSO.



Figure 8 describes the results of 20 individual models for the causal influences of NTA on boreal winter SST over the tropical Pacific region. Consistent with Figures 6 and 7a, most models exhibit nonsignificant NTA causal impacts on central equatorial Pacific winter SST, contributing to the weak response of ENSO to NTA. The impacts of NTA on central equatorial Pacific in reanalysis data (Figure 7b) suggest that NTA impacts are more active in the central equatorial Pacific rather than the eastern Pacific. However, this pattern is only observed in the model MPI-ESM1-2-XR (Figure 8). Several models (i.e., CAMS-CSM1-0, CMCC-CM2-VHR4, FGOALS-f3-L, and IPSL-CM6A-ATM-HR) suggest that NTA is likely to have causal impacts on eastern equatorial Pacific SST (Figure 8). These results suggest the inconsistency of models in modeling the impacts of NTA on the tropical Pacific.

Additional analyses showed the nonsignificant causal influences of NTA on the Atlantic-Pacific Walker circulation (Figure 9), consistent with the findings depicted in Figures 6 and 7. Specifically, there is weak response of tropical Pacific wind systems at 850-hPa (Figures 9a and b) and 250-hPa (Figures 9c and d) pressure levels to NTA. Substantial fluctuations in the causal impacts of NTA on ENSO are observed for the 41-year sliding probability (Figure 9e). The results indicate the unstable impacts of the NTA on ENSO, where there are periods (i.e., central years 1970-1978) of stronger NTA impacts compared to others. Reanalysis data suggests more steady impacts of NTA on ENSO in recent years (i.e., central years 1981-1994, except for the year 1986). The models' mean and the reanalysis show contradict results in several periods (i.e., central years 1971-1978). Although ENSO is still a dominant mode in the connection between tropical Pacific and Atlantic (Figures 1 and 5e), the results imply that NTA impacts are increased in recent years (Figure 9e).

## 4    Discussion and conclusions

The robust causal effects of ENSO on NTA (Figure 2) are in agreement with previous works (Alexander et al., 2002; García-Serrano et al., 2017; Saravanan and Chang, 2000; Tokinaga et al., 2019; Zhang et al., 2021). Further, the results described in Figures 3 and 5 support for the connection between ENSO and the north tropical Atlantic via the modulation of tropical Pacific-Atlantic Walker circulation (Saravanan and Chang, 2000) and the Hadley cell over the tropical Atlantic (Wang, 2004). Specifically, while ENSO impacts on tropical Atlantic zonal winds (Figures 3c, 5a and 5c) is caused by ENSO-induced tropical Pacific-Atlantic Walker circulation, ENSO impacts on tropical Atlantic meridional winds (Figures 3e, 5b and d) are triggered by changes in the local Hadley cell over the tropical Atlantic Ocean (Wang, 2004). In addition, the ENSO-induced response of surface temperature over the eastern tropical Pacific and northern South America (Figures 3a, b and 4) may play an important role in driving the NTA variations. During El Niño, this process is associated with the propagation of the equatorial Kelvin waves which contribute to the reduction in convection and the increase in SST over the north tropical Atlantic (Chang et al., 2006; Jiang and Li, 2019; Yang et al., 2021).

The lower amplitude of NTA for several events found in the models compared to reanalysis data (Figure 1) may lead to the weak response of ENSO and tropical Pacific to NTA (Figures 6-8). Weaker impacts of the NTA on ENSO in the model simulations compared to reanalysis data (Figure 6) suggest that the models may underestimate the strength of NTA. These

results show an agreement with the recent work (Jia et al., 2019; Yang et al., 2021) which suggested the bias in model simulations of the interaction between Pacific and Atlantic oceans.

The weak predictability power of NTA on ENSO events points to the important roles of the IOD and the IOB in the NTA-ENSO links, as discussed in previous works (Ha et al., 2017; Le et al., 2020; Le and Bae, 2019). The decline of NTA impacts on tropical Pacific SST during boreal fall and winter (Figures S1b and 7b) after peaking in boreal spring is

associated with the strengthening of the IOD impacts on ENSO (Le et al., 2020) and possible impacts of the IOB (Cai et al., 2019).

The results imply that better representation of NTA causal effects on ENSO in the models may improve the predictability of ENSO variations. Given the rising influence of the tropical Indian ocean (Abram et al., 2008; Le et al., 2020; Luo et al., 2012) and the Atlantic ocean (Cai et al., 2019; Li et al., 2016), further studies focusing on the causal connection between

NTA and ENSO under future warming environment are necessary.

**Acknowledgments**

We acknowledge the World Climate Research Programme, which through its Working Group on Coupled Modelling, coordinated and promoted CMIP6. We thank the climate modelling groups (listed in Table S1) for producing and making available their model output, the Earth System Grid Federation (ESGF) for archiving the data and providing access, and the

multiple funding agencies who support CMIP6 and ESGF. Thanh Le is supported by the National Research Foundation of Korea (NRF) grant funded by the Korea government (MSIT) (Grant No. 2021R1G1A1004389).

**Author Contributions**

T.L. designed the study, performed the data analysis, and wrote the manuscript. D.-H.B. contributed to the interpretation of results and the writing of the manuscript.

**Competing Interests**

The Authors declare no competing interests.

**Data Availability**

The data that support the findings of this study are openly available at the following website: https://esgf-node.llnl.gov/search/cmip6/.





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



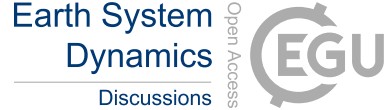

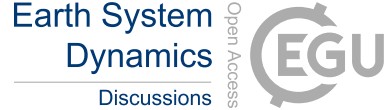



**Figure 1.** Variations of ENSO (a) and NTA (b) in the models (colored symbols) and reanalysis data (solid blue line) during the period 1950-2014. Strong positive ENSO and NTA events (exceeding +1.5 standard deviation $\sigma$) are marked by cyan shading. Red, dashed-red, and dashed-blue lines imply values of +3$\sigma$, +1.5$\sigma$ and -1.5$\sigma$, respectively. ENSO: El Niño–Southern Oscillation. NTA: North Tropical Atlantic.



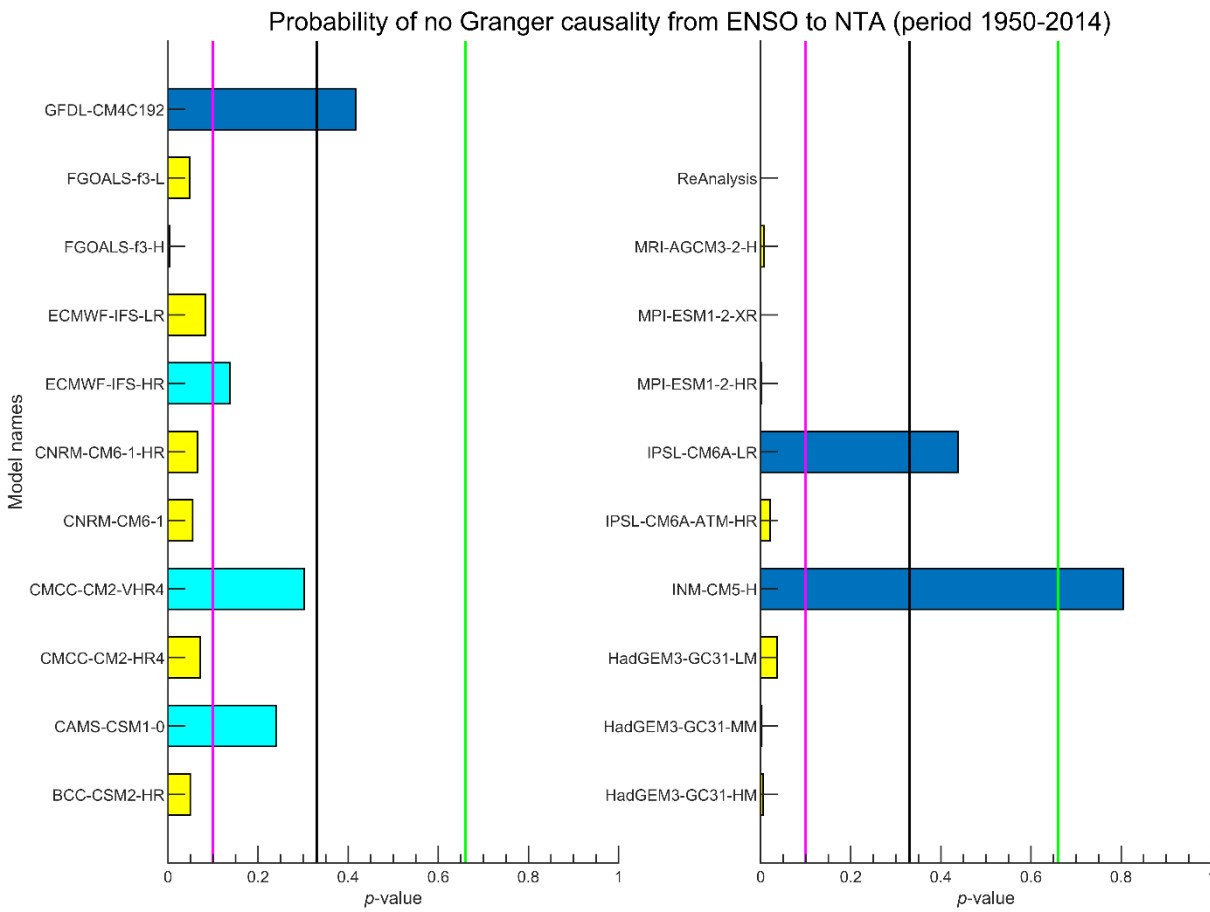


**Figure 2.** Probability of no Granger causal influences from ENSO to NTA in 20 individual models and reanalysis data for the period 1950-2014. Green, black, and magenta lines denote probability of 0.66, 0.33 and 0.1, respectively. ENSO: El Niño–Southern Oscillation. NTA: North Tropical Atlantic.





**Figure 3.** ENSO [D(*t*)JF(*t+1*)] effects on boreal spring [MAM(*t+1*)] north tropical Atlantic SST (a, b), surface zonal winds (c, d), and surface meridional winds (e, f) during the 1950–2014 period. The green and yellow contour lines specify *p* value = 0.05 and 0.1, respectively. Brown shades imply low probability for the absence of Granger causality. Stippling designates that more than 70% of models agree on the multi-model mean probability. For a given model, the agreement is defined when the discrepancy between that model's probability and the multi-model mean probability is less than one standard deviation of the multi-model mean probability. ENSO: El Niño–Southern Oscillation; SST: sea surface temperature; DJF: December–January–February; MAM: March-April-May.





**Figure 4.** ENSO [D($t$)JF($t+1$)] effects on boreal spring [MAM($t+1$)] north tropical Atlantic SST during the
1950–2014 period for 20 individual models. The green and yellow contour lines specify $p$ value = 0.05 and 0.1,
respectively. Brown shades imply low probability for the absence of Granger causality. ENSO: El Niño–Southern
Oscillation; SST: sea surface temperature; DJF: December–January–February; MAM: March-April-May.





(a) MODELS MEAN: ENSO - Spring UA850

(b) MODELS MEAN: ENSO - Spring va850

(c) MODELS MEAN: ENSO - Spring UA250

(d) MODELS MEAN: ENSO - Spring va250

0.1   0.2   0.3   0.4   0.5   0.6   0.7   0.8   0.9   1

(e) Sliding probability of no Granger causality from ENSO to NTA

$p$-value

Central year of 41-year sliding window

| | | | |
|---|---|---|---|
| BCC-CSM2-HR | ECMWF-IFS-HR | HadGEM3-GC31-MM | MPI-ESM1-2-XR |
| CAMS-CSM1-0 | ECMWF-IFS-LR | HadGEM3-GC31-LM | MRI-AGCM3-2-H |
| CMCC-CM2-HR4 | FGOALS-f3-H | INM-CM5-H | ReAnalysis |
| CMCC-CM2-VHR4 | FGOALS-f3-L | IPSL-CM6A-ATM-HR | Models mean |
| CNRM-CM6-1 | GFDL-CM4C192 | IPSL-CM6A-LR | |
| CNRM-CM6-1-HR | HadGEM3-GC31-HM | MPI-ESM1-2-HR | |



**Figure 5.** ENSO [D($t$)JF($t+1$)] effects on north tropical Atlantic 850-hPa zonal (a) and meridional (b) winds
during boreal spring [MAM($t+1$)]. ENSO [D($t$)JF($t+1$)] effects on north tropical Atlantic 250-hPa zonal (c) and
meridional (d) winds during boreal spring [MAM($t+1$)]. (e) 41-year sliding probability for the absence of
Granger causal impacts from ENSO to NTA in 20 individual models (thin lines), models mean (thick red line)
and reanalysis data (thick black line) for the period 1950-2014. Horizontal green, black and magenta lines denote
probability of 0.66, 0.33 and 0.05, respectively. ENSO: El Niño–Southern Oscillation. NTA: North Tropical
Atlantic. DJF: December–January–February; MAM: March-April-May.

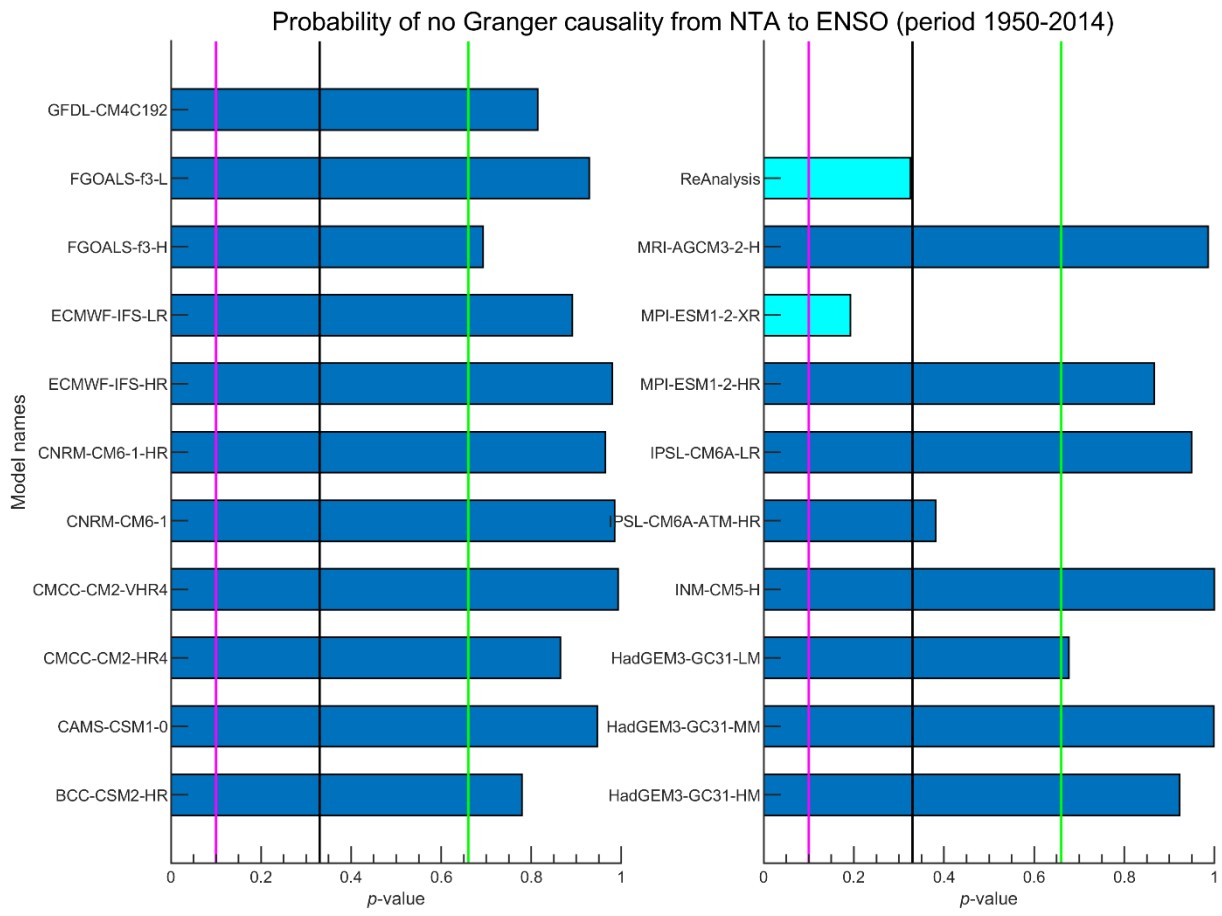

**Figure 6.** Probability of no Granger causal influences from NTA to ENSO in 20 individual models and reanalysis data for the period 1950-2014. Green, black, and magenta lines denote probability of 0.66, 0.33 and 0.1, respectively. ENSO: El Niño–Southern Oscillation. NTA: North Tropical Atlantic.






**Figure 7.** NTA [MAM(*t*)] effects on boreal winter [D(*t*)JF(*t+1*)] tropical Pacific SST (a, b), surface zonal winds (c, d), and surface meridional winds (e, f) during the 1950–2014 period. The yellow and cyan contour lines specify *p* value = 0.1 and 0.33, respectively. Brown shades imply low probability for the absence of Granger causality. Stippling designates that more than 70% of models agree on the multi-model mean probability. For a given model, the agreement is defined when the discrepancy between that model's probability and the multi-model mean probability is less than one standard deviation of the multi-model mean probability. NTA: North Tropical Atlantic; SST: sea surface temperature; DJF: December–January–February; MAM: March-April-May.







**Figure 8.** NTA [MAM($t$)] effects on boreal winter [D($t$)JF($t+1$)] tropical Pacific SST during the 1950–2014
period for 20 individual models. The yellow and cyan contour lines specify $p$ value = 0.1 and 0.33, respectively.
Brown shades imply low probability for the absence of Granger causality. NTA: North Tropical Atlantic; SST:
sea surface temperature; DJF: December–January–February; MAM: March-April-May.







**Figure 9.** NTA [MAM(*t*)] effects on tropical Pacific 850-hPa zonal (a) and meridional (b) winds during boreal winter [D(*t*)JF(*t+1*)]. NTA [MAM(*t*)] effects on tropical Pacific 250-hPa zonal (c) and meridional (d) winds during boreal winter [D(*t*)JF(*t+1*)]. (e) 41-year sliding probability for the absence of Granger causal impacts from NTA to ENSO of 20 individual models (thin lines), models mean (thick red line) and reanalysis data (thick black line) for the period 1950-2014. Horizontal green, black and magenta lines denote probability of 0.66, 0.33 and 0.05, respectively. ENSO: El Niño–Southern Oscillation. NTA: North Tropical Atlantic; SST: sea surface temperature; DJF: December–January–February; MAM: March-April-May.