# Peer review of "Causal interactions between ENSO and the North Tropical Atlantic"

_Earth System Dynamics, 2023_

## Author Comment (AC1)

**1 Response to Reviewer #1's comments**

The manuscript "Causal interactions between ENSO and the North Tropical Atlantic" by Thanh Le and Deg-Hyo Bae presents an assessment of the causal link between ENSO and NTA using both high-resolution climate models and reanalysis data. The idea of the manuscript is surely interesting and worthy of investigation, however it is difficult to read and the assessment needs to be strengthened from a statistical point of view. Below are my comments.

**Major comments**

1.1 The authors used monthly data but reconstructed indices at annual resolution (Figure 1). Then, they use Eq. S1 to assess the causality which considers yearly resolution time series. But then the authors discuss seasonal effects of ENSO and NTA. I would suggest the authors to better clarity how the analysis is carried out, unless it is not possible to replicate the results they obtained.

**Response:** Please see the supporting information and the references therein for additional information (Lines 16-26). We only focus on the peaking seasons of each climate mode. We do not focus on analyzing monthly data because of the different peaking seasons of each climate mode. For example, analyzing the relationship between ENSO in the decaying months (e.g., June) and NTA in the peaking months (e.g., April) is generally not of interest.

1.2 Another missing information is the number of lagged observations used for each model (p in Eq. S1) that is also strictly linked with my previous comment on the resolution. This can also affect the results since it depends on the resolution used and on the present/past observations. Which is the statistical threshold for significance?

**Response:** Please see the response in section 1.1. Please also see the supporting information for additional information on the degree of uncertainty (Lines 50-57).

1.3 The authors claim for a causal link between NTA and ENSO with directionality pointing from ENSO to NTA. The causality has been assessed, if I correctly understood, by using indices at annual resolution (although they used monthly data, see comment 1). Indices are firstly standardized and then the Granger causality has been evaluated, whose assessment is based on the p-value based IPCC-based recommendations. However, the Granger causality needs to be assessed with respect to a null-hypothesis that requires a statistical basis. Which is this statistical basis? Apart the p-value, did the authors performed some statistical tests based, as an example, on bootstraps procedures or random phases?

**Response:** Please see the supporting information and the references therein for additional information on the methods used in this work (Lines 47-57).

1.4 If the authors used yearly resolution, how the results are affected by the reduced size of samples (finite size effects)? It is the same for monthly resolution, thus a discussion and further additional tests are required to assess the robustness.

**Response:** Please see the response in section 1.1. Please also see the supporting information and the references therein for additional information.

## Minor comments

1.5 Line 23: remove the comma before "typically".

**Response:** We prefer to keep the original text.

1.6 Line 24: remove the comma after "spring" and change with "and it".

**Response:** We prefer to keep the original text.

1.7 Lines 66-67: should be the "first principal component"?

**Response:** The original text is correct.

1.8 The quality of all figures needs to be improved.

**Response:** We will provide high quality figures.

---

## Author Comment (AC2)

**2   Response to Reviewer #2's comments**

Review of  MS No.: esd-2023-1

Causal interactions between ENSO and the North Tropical Atlantic

Author(s): Thanh Le and Deg-Hyo Bae

The authors study causal interactions between the North Tropical Atlantic mode (NTA) and the El Niño–Southern Oscillation (ENSO), accounting for possible compounding effects of the Indian Ocean Dipole (IOD) and the Indian Ocean Basin mode (IOB), using a form of Granger causality (GC) analysis. This is an interesting topic potentially bringing important results.

Reading the paper, a number of technical questions emerged.

Sampling: The authors use monthly data, however they evaluate seasonal indices, leading to yearly data, i.e. they obtained the results using 65 samples. Questions:

**2.1   1. Model order- they mention two criteria, but do not specify which was finally used and which order of models was applied. Considering 65 samples, any order higher than 1 or 2 can be problematic, considering 4 variables in the models.**

**Response:** We should note that the model order and the number of 'variables' are independent variables. The order is related to the number of 'samples' as you mentioned. Please see the supporting information for additional information (Lines 39-46).

**2.2   2. The results are asymmetric, which might be interpretable as ENSO being more important climate mode. On the other hand, by data construction, the effect of ENSO on NTA was evaluated using a time delay of 3 months, while the other direction has inherent time lag 9 months.  Can this play any role? Would not be analysis using monthly data and different time lags interesting?**

**Response:** We only focus on the peaking seasons of each climate mode. We are not interested in analyzing monthly data because of the different peaking seasons of each climate mode. For example, analyzing the relationship between ENSO in the decaying months (e.g., June) and NTA in the peaking months (e.g., April) is generally not of interest. The time lags indeed play important roles and that is the point we want to test. In fact, despite long time lag, NTA impacts can still be
observed in reanalysis data and some models (Figures 7 and 8).

2.3   3. The results in Fig. 3 and other maps - are they obtained in the same way using yearly data,
just ENSO effect was evaluated on any grid-point separately? Is p-value mapped?
**Response:** Yes, they are the same approach.
The question 3 leads to:

2.4   4. If many tests are presented, is any correction to multiple testing considered? This can be
very critical esp. in Fig. 7, where a few significant spots could appear by chance. Not to
speak about very weak criteria taking as significant also values of p 0.1 -0.3. Good to remind
that typical conservative approach relies on $p<0.05$. I would conclude that no effect of NTA
on ENSO was detected, and very few models were able to reproduce the effect of ENSO on
NTA, observed in the reanalysis data. Btw. in Fig. 2 the results for the reanalysis data is not
visible because of small p-value? This should be mentioned in the caption and the value
should be written.
**Response:** We followed the approach suggested in previous work to discuss the degree of
uncertainty (Lines 50-57, supporting information). Fundamentally, the results of causal impact or
no causal impact are equally important and both results should be reported.

2.5   5. Another remark to significance levels - if they are obtained from an analytic expression,
there are always some data requirements. I propose to add some computational statistics such
as surrogate data or bootstrap, to avoid false significance.
**Response:** Please see the response in section 2.4. In our opinion, the study is well designed.

2.6   6. Multiplicity correction applies also for sliding window results, e.g. Fig. 9e.
**Response:** Please see the response in section 2.4.

2.7   7. What would be the results without accounting for IOD and IOB?
**Response:** We aim to provide information close to real-world teleconnections. Hence, accounting
the impacts of IOD and IOB are necessary. For this reason, we have not tried to produce analyses
excluding these two modes.

2.8   8. If p-values is mapped in Figs. 3 and similar, I would not talk about the causal effect.

p-value gives the reliability of inference (of causality in this case) which is not generally equivalent to (physical) causal effect of the cause on the effect variable.

**Response:** Please see the response in section 2.4. The results shown in Figures 3, 5 and 7 support the causal effects (i.e., the modulation of atmospheric circulation related to ENSO and NTA).